# The Effect of Heat Treatment on the Corrosion Resistance of Fe-Based Amorphous Alloy Coating Prepared by High Velocity Oxygen Fuel Method

**DOI:** 10.3390/ma14247818

**Published:** 2021-12-17

**Authors:** Chun-Ying Lee, Hung-Hua Sheu, Leu-Wen Tsay, Po-Sen Hsiao, Tzu-Jing Lin, Hung-Bin Lee

**Affiliations:** 1Graduate Institute of Manufacturing Technology, National Taipei University of Technology, Taipei 106, Taiwan; leech@ntut.edu.tw; 2Department of Chemical & Materials Engineering, Chung Cheng Institute of Technology, National Defense University, Taoyuan 335, Taiwan; shhccit@gmail.com; 3System Engineering and Technology Program, National Yang Ming Chiao Tung University, Hsin-Chu 300, Taiwan; 4Department of Optoelectronics and Materials Technology & Center of Excellence for Ocean Engineering, National Taiwan Ocean University, Keelung 202, Taiwan; b0186@mail.ntou.edu.tw (L.-W.T.); portershiau@gmail.com (P.-S.H.); abc02025926@gmail.com (T.-J.L.)

**Keywords:** amorphous structure, corrosion, heat treatment, HVOF

## Abstract

In this study, Fe_40_Cr_19_Mo_18_C_15_B_8_ amorphous coatings were prepared using high velocity oxygen fuel (HVOF) technology. Different temperatures were used in the heat treatment (600 °C, 650 °C, and 700 °C) and the annealed coatings were analyzed by DSC, SEM, TEM, and XRD. XRD and DSC results showed that the coating started to form a crystalline structure after annealing at 650 °C. From the SEM observation, it can be found that when the annealing temperature of the Fe-based amorphous alloy coating reached 700 °C, the surface morphology of the coating became relatively flat. TEM observation showed that when the annealing temperature of the Fe-based amorphous alloy coating was 700 °C, crystal grains in the coating recrystallized with a grain size of 5–20 nm. SAED analysis showed that the precipitated carbide phase was M_23_C_6_ phase with different crystal orientations (M = Fe, Cr, Mo). Finally, the corrosion polarization curve showed that the corrosion current density of the coating after annealing only increased by 9.13 μA/cm^2^, which indicated that the coating after annealing treatment still had excellent corrosion resistance. It also proved that the Fe-based amorphous alloy coating can be used in high-temperature environments. XPS analysis showed that after annealing FeO and Fe_2_O_3_ oxide components increased, and the formation of a large number of crystals in the coating resulted in a decrease in corrosion resistance.

## 1. Introduction

The method of manufacturing amorphous metals involves melting the metals down and applying a rapid cooling rate, so that the metal atoms do not have chance to form an orderly arrangement during the condensation process and form an amorphous state. Amorphous metals are free of grains and grain boundaries, which results in high-strength mechanical properties [1], high corrosion resistance [2], and excellent soft magnetization [3]. Among amorphous metals, the Fe-based amorphous alloy has wider industrial application due to its relatively low cost—in marine contexts, industry, and medicine, amongst other fields [4,5,6]. Recently, due to the high hardness and corrosion resistance of Fe-based amorphous alloy, its method of manufacture has been promoted as one of the methods to protect metal substrates, such as 304SS [7], 316SS [8], and EHC [9]. The preparation of Fe-based amorphous alloy coatings is mainly completed with thermal spraying. The technologies of thermal spraying include plasma spraying [10], arc spraying [11], high-speed flame spraying [12]. The coating prepared by HVOF has a low oxygen content and low porosity [13]. Zhang et al. [14] used HVOF technology to prepare an Fe_49.7_Cr_18_Mn_1.9_Mo_7.4_W_1.6_B_15.2_C_3.8_Si_2.4_ coating. There was no obvious corrosion on the coating surface after 42 days of a salt spray test. However, after observing corrosion behavior through a polarization test, it was found that corrosion occurred along the periphery of pores of the coating. According to ToF-SIMS analysis, it was found that there was a lack of chromium around the pores, leading to the spread of corrosion. Lee et al. [15] prepared Fe_40_Cr_19_Mo_18_C_15_B_8_ alloy coatings using the HVOF method and compared them with 304SS and 316SS. They found that although the corrosion current density of 304SS and 316SS is lower than that of Fe-based amorphous alloy, the width of the passivation zone of stainless steel is smaller than that of the latter. This shows that the corrosion resistance of Fe-based amorphous alloy comes from its passivation protection. The corrosion mechanism of the Fe-based amorphous alloy coating was observed by polarization test, and the defects of the coating were found to come from cracks and recrystallization during thermal spraying under TEM observation.

Amorphous metals are metastable materials, and recrystallization will occur during the annealing process. However, the effect of recrystallization on the amorphous metal does not completely lead to the reduction of its corrosion resistance. Wu et al. [16] and Yang et al. [17] indicated that when the annealing temperature increases the amount of recrystallized crystalline phase in the amorphous coating will increase, and these crystalline phases will form poor-chromium and poor-molybdenum regions, leading to a decrease in the corrosion resistance of the coating. Liang et al. [18] and Belkhaouda et al. [19] found that the annealing treatment of FeCrMoWCBY amorphous alloy at the glass transition temperature will reduce the passivation current density (i_pass_) of the coating, indicating that the annealing treatment contributes to the formation of a stable passivation layer and improves the corrosion resistance of the coating. The amorphous coating has the best corrosion resistance when the annealed temperature is 0.7 T_g_ and heated for 30 min. Kang et al. [20] reported that the Fe_48_Cr_15_Mo_14_C_15_B_6_Y_2_ amorphous alloy coatings are annealed at 500 °C and 700 °C, respectively, and found that after annealing the coating will undergo structural relaxation and recrystallization. Structural relaxation may lead to the formation of a denser passivation layer and increase the corrosion resistance of the coating. With annealing at 700 °C, excessive crystalline phases will be formed in the coating, which will reduce the corrosion resistance of the coating.

Research into annealing treatments has not been entirely negative with respect to corrosion resistance and wear resistance in amorphous materials. In the studies of Koga et al. [21] and Liu et al. [22], it was found that after annealing treatment, the corrosion resistance and wear resistance of the coating will show a contradiction, that is, the coating will form a (Fe, Cr)_2_B precipitate phase, which in turn increases the hardness of the coating and improves wear resistance. In addition, controlling the annealing temperature of the coating can achieve the best wear resistance and corrosion resistance. In most of the literature on the effect of annealing treatments on Fe-based amorphous alloy, the research results show that after the annealing treatment of the amorphous coating, the increase of the crystalline phase of the coating can be observed under XRD and TEM analysis, although the defects formed by crystal precipitation will cause the corrosion resistance of the coating to decrease. However, when the hardness of the alloy coating is increased at the same time, the wear resistance of the coating will be further improved [23,24,25]. Therefore, in this study, the Fe_40_Cr_19_Mo_18_C_15_B_8_ amorphous coating was annealed at different temperatures, and XRD, SEM, and TEM were used to study the influence of the amorphous coating at different annealing temperatures.

## 2. Materials and Methods

### 2.1. Preparation of the Fe-Based Amorphous Alloy Coating

First remove the grease on the surface of the substrate with alcohol, and then perform surface sandblasting to strengthen the surface of the substrate and increase the roughness to Ra 50–80 μm to increase the adhesion of the thermal spray coating. The metallic powders of Fe-based amorphous alloy powders with particle sizes between 30–50 μm were baked in an oven at 80 °C for one hour to remove the water content. The parameters of the HVOF process were set as follows: the propane flow rate at 48 F.M.R., oxygen flow rate at 29 F.M.R., powder feeding rate at 35 g/min, and moving speed of spray gun at 800 mm/sec [15]. After the HVOF process, annealing treatment of the Fe-based amorphous alloy coatings were carried out with different respective temperatures (550 °C, 650 °C, and 700 °C) for 1 h in a high-temperature furnace.

### 2.2. Characterization Analysis of the Fe-Based Amorphous Alloy Coating

An X-ray diffractometer (D8 SSS, BRUKER Co. Ltd., Billerica, MA, USA) with Cu Kα radiation (λ = 0.15405 nm) was employed to identify the phase composition of the sprayed coating after heat treatment by setting a scanning rate of 0.02°/s and a scanning range from 20°–100°. For the examinations of the surface and cross-sectional morphology, a scanning electron microscope (Hitachi SEM S-3400N, Hitachi High-Tech. Co., Ltd., Tokyo, Japan) and a transmission electron microscope (JEOL JEM-2100F, JEOL Ltd., Tokyo, Japan) were used, respectively. To analyze the phase transformation of the prepared coating, the thermogravimetry–differential scanning calorimetry (TG–DSC) (Netzsch 404F3, NETZSCH-Gerätebau GmbH, Selb, Germany) measurements were performed at heating rates of 10 °C/min from room temperature to 1350 °C (1623 K) in an N_2_ atmosphere. As for the electrochemical characterization of the coating, the potentiodynamic polarization curves were measured in a standard three-electrode cell system and carried out with a Zahner Zennium (E 41100, ZAHNER-elektrik GmbH & Co., Kronach, Germany). In the testing configuration, the Fe-based amorphous alloy coating specimen, a saturated calomel electrode (SCE), and a platinum plate were used as the working electrode (WE), the reference electrode, and the counter electrode, respectively. The measurement of the potentiodynamic polarization curve in a 3.5 wt.% NaCl solution at room temperature within a potential range between −1.0 V and 1.5 V and with a scanning rate of 0.5 mV s^−1^ was started after 20 min pre-immersion.

## 3. Results

### 3.1. Morphologies of Fe-Based Amorphous Alloy

The Fe-based amorphous alloy powder is formed by high-temperature melting and rapid atomization. Figure 1a presents an SEM image of Fe-based amorphous alloy coating powder. It can be observed that most of the Fe-based amorphous alloy powder has a nearly spherical structure, and the particle size of the powder is about 30–50 μm. The results of the SEM-EDS analysis of the powder are shown in Table 1. The detected oxygen content may be caused by the oxidation of the Fe-based amorphous alloy powder during the atomization process. The surface morphology of the HVOF spray coating is shown in Figure 1b. It can be seen that the coating surface is rugged and contains many particle shapes of different sizes. The large particles (30–50 μm) are considered unmelted particles, while the finer particles (~10 μm) are caused by powder melting and splashing during spraying. The SEM cross-sectional image of the Fe-based amorphous alloy coating is shown in Figure 1c. From the figure, it can be observed that the surface of the coating is uneven and that there are some dimples. The thickness of the coating is about 270 μm and it contains many pores (indicated by the white arrow in Figure 1c). From the SEM-EDS analysis results in Table 1, it is known that the Fe-based amorphous alloy powder will have a higher oxygen content after thermal spraying. Figure 1d shows the TEM image of the Fe-based amorphous alloy coating. It can be seen that some areas of the coating have defects, such as grains and cracks (the red circles in Figure 1d). The grain size is about 20 nm, and it is judged that a little recrystallized grain is formed due to overheating during the spraying process. Moreover, during cooling and shrinking micro-cracks are formed due to the presence of grains in the coating [15].

### 3.2. Microstructure of Fe-Based Amorphous Alloy Coatings after Annealing

Figure 2a shows the X-ray diffraction pattern of the Fe-based amorphous alloy coating after annealing. The Fe-based amorphous alloy coating still presents a broad diffraction peak, indicating that the coating retains an amorphous structure after annealing at 600 °C, while the amorphous alloy coating begins to recrystallize into a polycrystalline structure when the annealing temperature increases to 650 °C, in which time the phase of the Fe-based amorphous alloy coating is composed of an α-Fe and M_23_(C, B)_6_ compound. The coating is completely transformed into a polycrystalline structure when the annealing temperature increases to 700 °C, and the X-ray diffraction pattern presents a phase structure for the Fe-based amorphous alloy coating composed of α-Fe and M(C,B) compounds, such as M_23_(C, B)_6_, M_3_(C, B)_2_, M(C, B)_2_, and M(C, B) [26]. The results of the analysis of DSC are shown in Figure 2b and Table 2. The recrystallization exothermic peaks both occurred in the samples with and without annealing at 600 °C. Kanno et al. [27] indicated that the amorphous structure of the alloy is stable when the value of T_g_/T_m_ is near 2/3. In this study, the value of T_g_/T_m_ of the iron-based amorphous alloy coating slightly increased from 0.651 to 0.658. It was not surprising that the amorphous rate of the material was almost unchanged after annealing at 600 °C. When the annealing temperature is increased to 650 °C and 700 °C, there is no recrystallization exothermic peak in the DSC curve. This result is consistent with the XRD diffraction result; the amorphous structure is completely transformed into a polycrystalline structure, so there will be no exothermic reaction of recrystallization on the DSC curve. The SEM morphologies of the Fe-based amorphous alloy coatings after annealing are shown in Figure 3. When the annealing temperature is 600 °C or 650 °C, a granular structure can be observed on the surface of the Fe-based amorphous alloy coating. Although the surface morphology of the coating becomes relatively flat, there are many micropores formed at the same time when the annealing temperature is increased to 700 °C. These micropores may reduce the corrosion resistance of the coating (the white arrow in Figure 3c). Moreover, in the SEM-EDS chemical composition analysis of the coating (Table 3), it can be seen that the content of oxygen increases and the content of molybdenum decreases after annealing.

### 3.3. Corrosion Resistance of Fe-Based Amorphous Alloy Coatings after Annealing

Figure 4 shows the polarization curves of the Fe-based amorphous alloy coating carried out at different annealing temperatures in 3.5 wt.% NaCl solution; the values of E_corr_, i_corr_, i_pass_ and E_pit_ are also shown in Table 4. When the Fe-based amorphous alloy coating annealed at 600 °C, the value of E_corr_ increased from –0.78 V_SCE_ (without annealing) to –0.51 V_SCE_. From the values of i_pass_ and E_pit_, it can be seen that the corrosion resistance of the Fe-based amorphous alloy coating decreased with an increase in the crystallinity of the coating, resulting in increased values of passive current density (i_pass_) and pitting potential (E_pit_). At different annealing temperatures, the value of E_corr_ for the Fe-based amorphous alloy coating will decrease with an increase in annealing temperature. The value of i_corr_ for the Fe-based amorphous alloy coatings annealed at 600 °C, 650 °C, and 700 °C is 33.93, 36.35, and 38.31 μA/cm^2^, respectively. All the values of i_corr_ for the Fe-based amorphous alloy coatings after annealing are worse than for the unheated coating. However, the difference in corrosion current density between the coating that has not been annealed and the coating that has been annealed at 700 °C is only 9.13 μA/cm^2^, indicating that the Fe-based amorphous alloy coating still has excellent corrosion resistance in high-temperature environments. From the XRD and DSC results, it can be seen that the annealing treatment leads to an increase in the corrosion current density of the Fe-based amorphous alloy coating. The reason is that a large number of crystalline phases are produced by recrystallization processes which give rise to chromium-poor and molybdenum-poor areas within coatings, thereby reducing the corrosion resistance of coatings [16,17,28].

## 4. Discussion

### 4.1. Calculation of Activation Energy for Recrystallization of Fe-Based Amorphous Alloys

Figure 5 illustrates the effect of heating rates (10 °C/min, 20 °C/min, 30 °C/min, 40 °C/min) on the DSC recrystallization peaks of FeCrMoCB amorphous powders. The faster the heating rate, the higher the temperature of the exothermic peak of recrystallization. The T_p_ of FeCrMoCB amorphous powders at different heating rates is 693.38, 701.21, 704.25, and 711.07 °C, respectively. The activation energy for recrystallization of Fe-based amorphous alloys are calculated using Kissiger’s equation [29,30]. The calculated recrystallization activation energy of FeCrMoCB amorphous powders is equal to 276.58 kJ/mol (Figure 5b) and the measured activation energy is close to the recrystallization activation energy of FeCrMoCB amorphous powders (385 kJ/mol) [31] and FeCrMoYCB amorphous powders (369 kJ/mol) [32]. The difference in the activation energy of recrystallization is caused by a difference in the chemical composition of the amorphous powders.

### 4.2. Microstructure Analysis of Fe-Based Amorphous Alloy Coatings after Annealing

Figure 6 presents TEM and SAED images of Fe-based amorphous alloy coatings after annealing at different temperatures. The microstructure of coatings without annealing is amorphous (Figure 6a), as are the structures of coatings annealed at 600 °C (Figure 6b). The result of SAED is in agreement with XRD patterns. For the coating annealed at 700 °C, the TEM image shows the formation of new grains from an amorphous structure during the recrystallization process (Figure 6c), and this result is also in agreement with XRD patterns (Figure 2a). The above phenomenon is similar to phenomena reported by Yang et al. [17] and Fu et al. [33]. After analysis, it was determined that, in addition to the iron phase, these crystalline phases also contained compound phases, such as M_23_C_6_ and M_7_C_3_ (M = Fe, Cr, Mo). After further magnification and observation, the grain size of these crystalline phases was about 5–20 nm. After TEM-EDS composition analysis, it was found that the grains formed at the annealing temperature of 700 °C had a high carbon content (43.02–51.76 at.%). This is because a large amount of carbide phase will have formed in the coating under the effect of the higher annealing temperature. In addition, during high-temperature annealing treatment, oxides will also be formed in the coating, which in turn increases the oxygen content from 2.40 at.% in the initial coating to 16.76 at.%. This shows that the deterioration in the corrosion resistance of the coating after the annealing treatment is caused by the formation of crystal precipitates and oxides during the high-temperature annealing treatment (Figure 6e,f and Table 5).

### 4.3. Corrosion Behavior of Fe-Based Amorphous Alloy Coatings after Annealing

By comparing the polarization curves of Fe-based amorphous alloy coatings with annealing (crystalline structure) and without annealing (amorphous structure), it was observed that the protective performance of the passivation film of the crystalline coating was reduced. Therefore, XPS was used to analyze the composition and chemical state of the elements in the passive film of the amorphous and annealed coatings. Figure 7a,c,e, respectively, presents the XPS spectrum corresponding to the iron core level spectra, chromium core level spectra, and molybdenum core level spectra of Fe-based amorphous alloy coatings without annealing. The binding energy of Fe2p_3/2_ and Fe2p_1/2_ orbitals in the iron-based amorphous coating without annealing is composed of Fe^0^ = 720.0 eV, 707.0 eV and Fe(II) = 723.3 eV, 709.3 eV [34,35,36]; the binding energy of Cr2p_3/2_ and Cr2p_1/2_ orbitals in the Fe-based amorphous alloy coating without annealing is composed of Cr^0^ = 583.5 eV, 574.1 eV and Cr(III) = 586.1 eV, 575.9 eV [33,34]; and the binding energy of molybdenum is composed of Mo^0^ = 227.8 eV; Mo(IV) = 231.0 eV [37]. Figure 7b,d,f, respectively, presents the XPS spectrum corresponding to the iron core level spectra, chromium core level spectra, and molybdenum core level spectra of Fe-based amorphous alloy coatings annealed at 700 °C. The binding energy of Fe2p_3/2_ and Fe2p_1/2_ orbitals in the based amorphous alloy coating annealed at 700 °C is composed of Fe^0^ = 720.0 eV, 707.0 eV and Fe(II) = 723.3 eV, 709.3 eV, and in addition Fe(III) = 711.6 eV [34,35,36]. The binding energy of Cr2p_3/2_ and Cr2p_1/2_ orbitals in the Fe-based amorphous alloy coating annealed at 700 °C is in addition composed of Cr^0^ = 583.5 eV, 574.1 eV and Cr(III) = 586.1 eV, 575.9 eV, also Cr(VI) = 577.7 eV [34,35]. The binding energy of molybdenum is in addition composed of Mo^0^ = 227.8 eV; Mo(IV) = 231.0 eV; there is also Mo(VI) = 232.3 eV [36]. The Fe(III), Cr(VI), and Mo(VI) which exist within the Fe-based amorphous alloy coating after annealing at 700 °C can be attributed to the formation of FeO, Fe_2_O_3_, Cr_2_O_3_, and MoO_3_ during corrosion (the first two are porous structures and the latter two are dense structures). However, the overall corrosion resistance of the coating is reduced, which is attributed to the formation of a large amount of carbon compound phases observed in TEM in Figure 6. Ha et al. [38] and Lee et al. [39] proved through experimental results that carbon compounds absorb surrounding Cr and Mo components when they are formed, which in turn leads to a lack of protective elements, such as Cr and Mo, in some areas, so defects will form around these crystals and be corroded.

## 5. Conclusions


The Fe-based amorphous alloy coating will start to undergo recrystallization and carbide phase precipitation reactions when the annealing temperature is above 650 °C.The results of the corrosion polarization curve show that the corrosion current density of the coating after annealing only increased by 9.13 μA/cm^2^, which indicates that the coating after annealing treatment still has excellent corrosion resistance. It also proves that the Fe-based amorphous alloy coating can be used in high-temperature environments.TEM observation results show that the Fe-based amorphous alloy coating still maintains an amorphous structure after annealing at 600 °C. When the annealing temperature is increased to 700 °C, a carbide phase will form in the amorphous coating. SAED analysis shows that the carbide phase is composed of M_23_C_6_ crystals with different crystal lattice directions.According to XPS analysis, the corrosion resistance is reduced due to the formation of a large amount of iron oxides in the coating after annealing at 700 °C.


## Figures and Tables

**Figure 1 materials-14-07818-f001:**
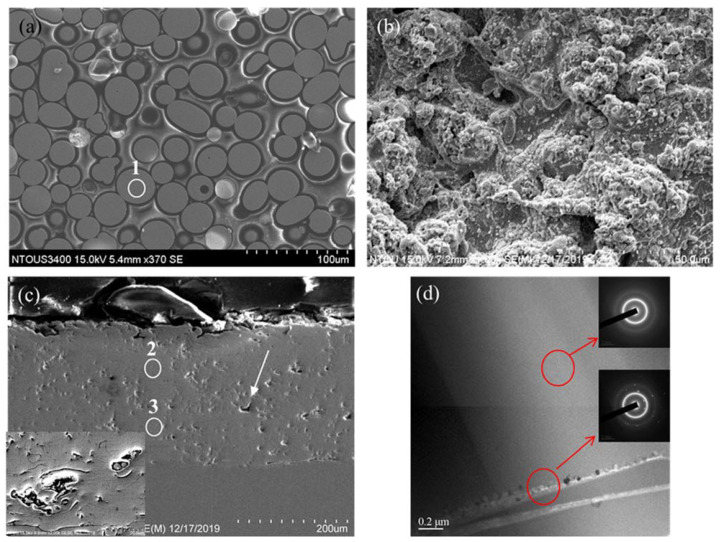
SEM images of Fe-based amorphous alloy coating before annealing: (**a**) initial powders, (**b**) surface of coating, (**c**) cross-section of coating, (**d**) TEM image of coating.

**Figure 2 materials-14-07818-f002:**
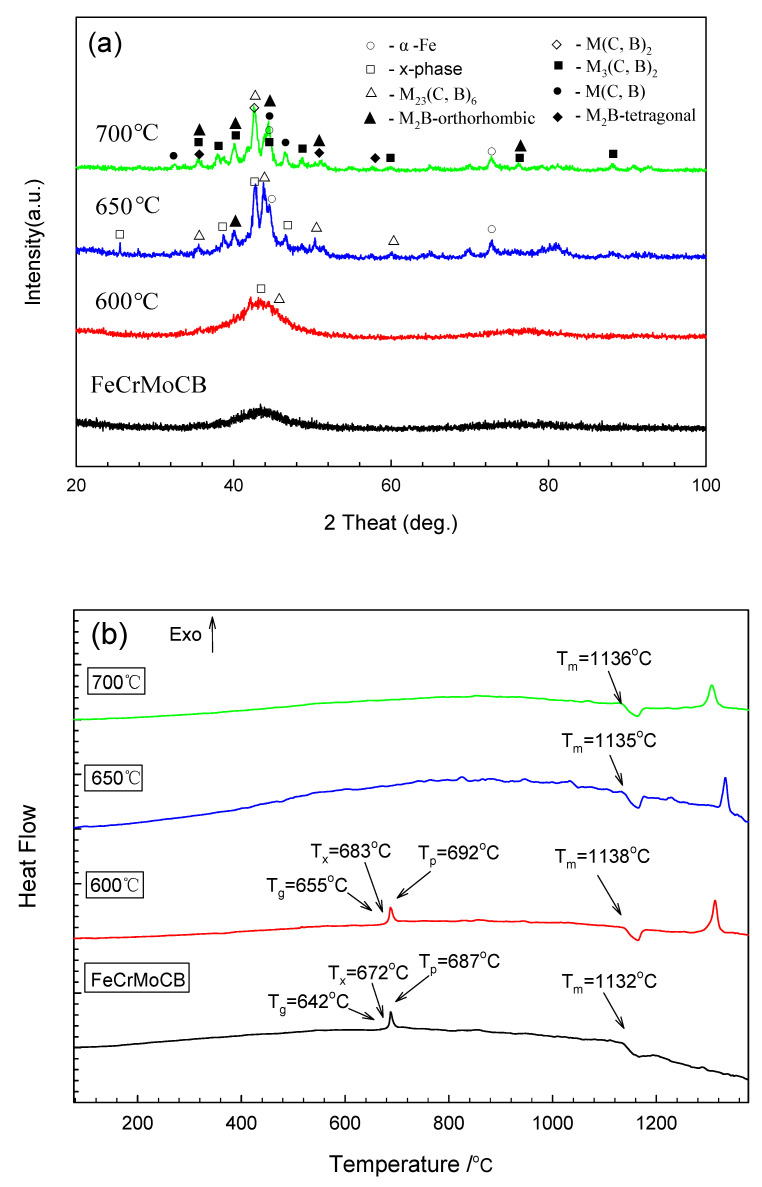
Measured results of the sprayed coating: (**a**) XRD spectrum, (**b**) DSC curves.

**Figure 3 materials-14-07818-f003:**
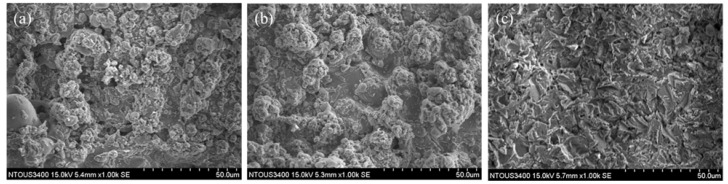
SEM morphologies of the Fe-based amorphous alloy coating after annealing at different temperatures: (**a**) 600 °C, (**b**) 650 °C, (**c**) 700 °C.

**Figure 4 materials-14-07818-f004:**
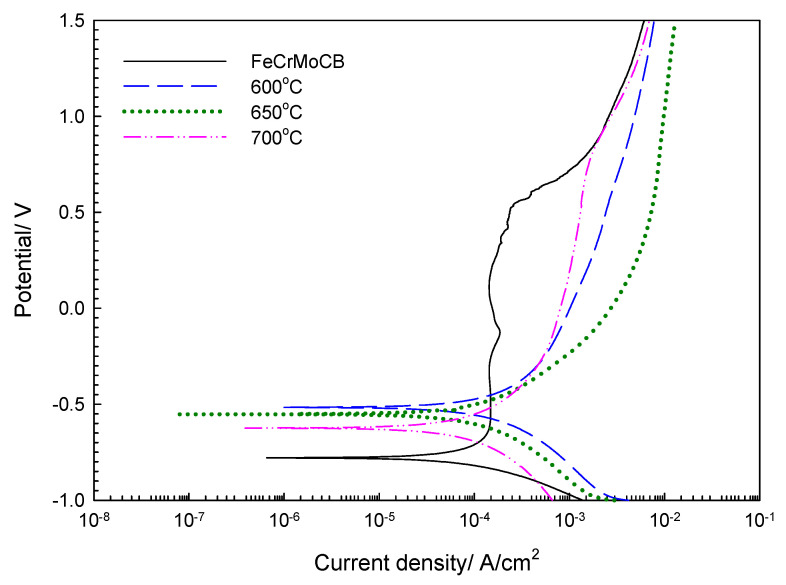
Polarization curves of Fe-based amorphous alloy coatings carried out with different annealing temperature.

**Figure 5 materials-14-07818-f005:**
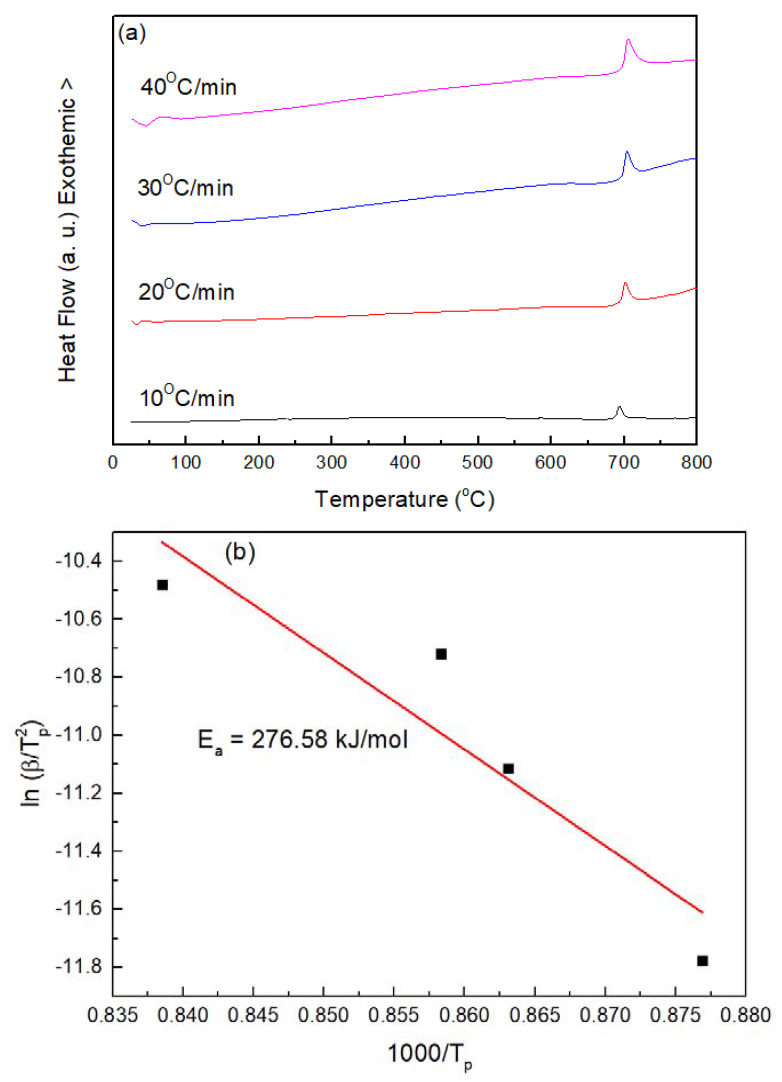
(**a**) DSC curves of Fe-based amorphous alloy coatings using heating rates of 10, 20, 30, and 40 °C/min. (**b**) Kissinger plot for the recrystallization exothermic peaks of Fe-based amorphous alloy coatings.

**Figure 6 materials-14-07818-f006:**
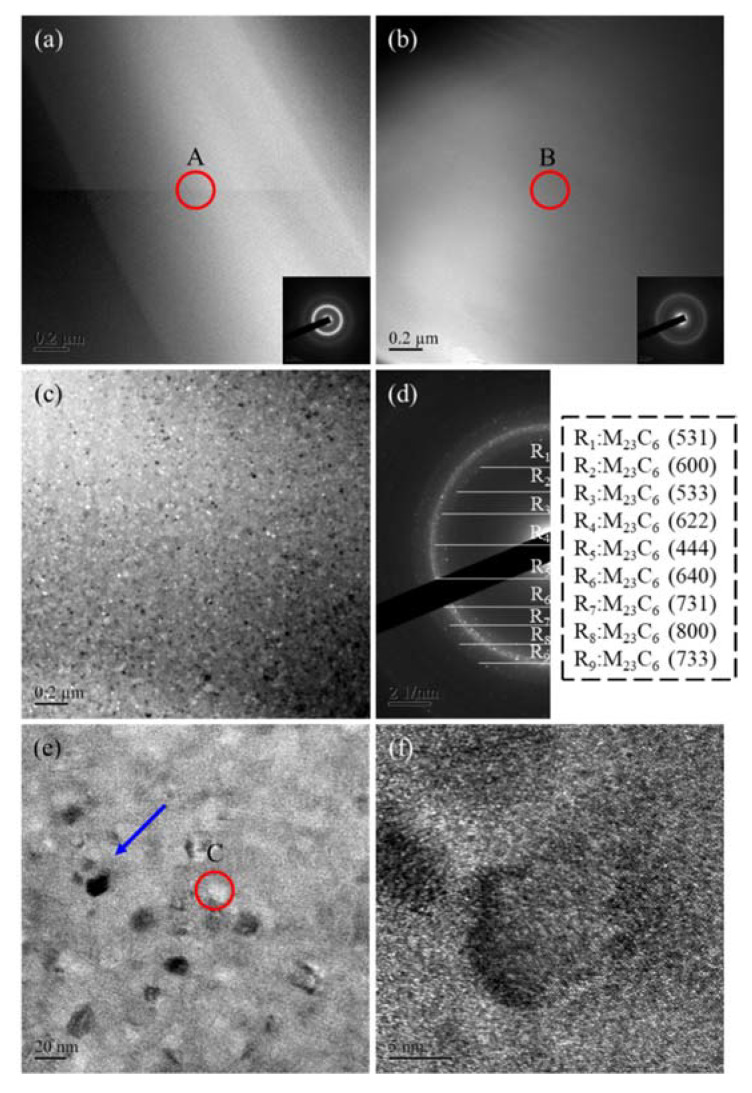
TEM images of coatings after annealing: (**a**) initial coating, (**b**) after annealing at 600 °C, (**c**) after annealing at 700 °C. (**d**) SAED image of coating after annealing at 700 °C. (**e**,**f**) Magnification photos of coating after annealing at 700 °C.

**Figure 7 materials-14-07818-f007:**
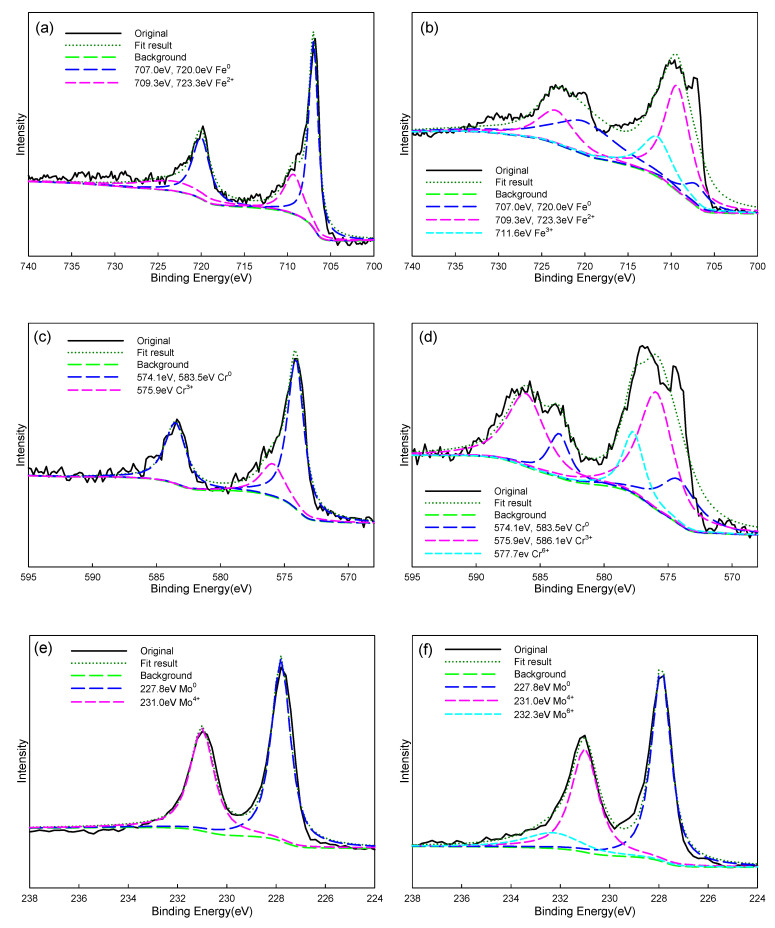
XPS spectra of passive films on the coatings initially (**a**,**c**,**e**) and after annealing at 700 °C (**b**,**d**,**f**).

**Table 1 materials-14-07818-t001:** SEM-EDS chemical composition analysis of Fe-based amorphous alloy coating before annealing (the analyzed positions are indicated by the white circles in Figure 1a,c).

	Elements (At.%)	Fe	Cr	Mo	C	O	Total
Position							
Raw material particles							
1		41.58	18.49	17.33	17.93	4.67	100.00
As-sprayed coating							
2		39.98	19.59	18.99	15.13	6.31	100.00
3		40.98	19.22	18.28	14.92	6.60	100.00
Average		40.48 ± 0.001	19.41 ± 0.001	18.64 ± 0.001	15.03 ± 0.001	6.46 ± 0.001	100.00

**Table 2 materials-14-07818-t002:** Related temperature parameters calculated from DSC curves.

	T_g_ (K)	T_x_ (K)	ΔT_x_ (K)	T_m_ (K)	T_g_/T_m_	T_p_ (K)
FeCrMoCB	915	945	30	1405	0.651	960
600 °C annealed	928	956	28	1411	0.658	965
650 °C annealed	-	-	-	1408	-	-
700 °C annealed	-	-	-	1409	-	-

**Table 3 materials-14-07818-t003:** SEM-EDS chemical composition analysis of the Fe-based amorphous alloy coating after different annealed temperature.

At.%	Fe	Cr	Mo	C	O	Total
As-sprayed	40.48	19.41	18.64	15.03	6.46	100.00
600 °C annealed	23.40	19.05	15.29	11.99	30.27	100.00
650 °C annealed	20.98	22.68	6.02	13.10	37.21	100.00
700 °C annealed	20.54	23.58	5.71	7.40	42.77	100.00

**Table 4 materials-14-07818-t004:** Corrosion potential and corrosion current density measured after annealing of Fe-based amorphous alloy coatings.

	E_corr_ (V_SCE_)	i_corr_ (A/cm^2^)	i_pass_ (A/cm^2^)	E_pit_
FeCrMoCB	−0.78	29.18 × 10^−^^6^	1.11 × 10^−^^4^	0.545
600 °C annealed	−0.51	33.93 × 10^−^^6^	28.61 × 10^−^^4^	0.642
650 °C annealed	−0.55	36.35 × 10^−^^6^	80.82 × 10^−^^4^	0.693
700 °C annealed	−0.63	38.31 × 10^−^^6^	10.46 × 10^−^^4^	0.831

**Table 5 materials-14-07818-t005:** Chemical composition analysis of coating after annealing at 700 °C using TEM-EDS (in red circled area).

At.%	Fe	Cr	Mo	C	O	Total
A	47.29	22.01	18.55	9.75	2.40	100.00
B	33.30	15.74	15.35	24.81	10.80	100.00
C	29.62	13.62	15.98	27.18	13.60	100.00

## Data Availability

All the data is available within the manuscript.

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
