# Peer review of "The Effect of Heat Treatment on the Corrosion Resistance of Fe-Based Amorphous Alloy Coating Prepared by High Velocity Oxygen Fuel Method"

_materials, 2021, doi:10.3390/ma14247818_

Round 1
Reviewer 1 Report
There is a lot of unnecessary hyphenation.
It’s better to check English, especially in the introduction.
There is a lot of orthographical variants
304SS and 304 stain-less steel
316SS and 316 stain-less steel
oC and ℃
Fe-based amorphous, iron-based amorphous, and iron-based amorphous alloy
Is it a field emission scanning electron microscope? (109
I think pore is proper to indicate hole. (136)
I couldn’t find cracks in Fig 1(c) and pores and cracks in Fig. 1 (d). Can you put some arrows to show them clearly?
There was a mistake regarding Tg/Tm. I think Tg/Tm in FeCrMoCB is 0.567 not 0.65.
And I’m afraid that Kelvin is used to calculating Tg/Tm in [27], not Celsius. (162)
I think it’s better to add information on the as-sprayed coating in Table 3.
I think it’s better to change the line of 650 ℃ in Fig. 4 and the lines of Fit result in Fig. 7.
Questions
Are there any changes in pores and cracks in coating annealed at 700℃? Does it affect corrosion properties?
Why the carbon amount of point C in Fig. 6 (e) is much higher than the composition shown in Fig. 6 (d)?
Author Response
Response to the question in the file.
Thank you for your advice.

Reviewer 2 Report
The presented manuscript includes the study of the effect of heat treatment on the corrosion resistance of iron-based amorphous alloy coatings prepared by high velocity oxygen fuel method.
The manuscript is well written and structured, but, a couple of corrections and weak points could be mentioned:
Q1. I can suggest to rename the title of the 2d section of the manuscript to “Materials and Methods” and dividing it into subsections, like such draft examples: 2.1 Materials, 2.2 Sample preparation/modification/treatment, 2.3 Materials characterization
Q2. There is a lack of information about the corrosion tests results in the main part of the manuscript. This is the weakest part of the manuscript. For such types of alloys, other parameters are usually measured. See 10.1038/s41597-021-00840-y . Icorr is very low for such types of alloys and is not so vital parameter. Please add as minimum OCP and Epit, ipass, etc for PD results.
Q3. “Cr6+” incorrectly written. Charge, ion, and valence have different designations in chemistry. Ion Cr6+ does not exist. So, I can suggest writing in Cr(VI) style and correct others to keep the same style.
Q4. Please add standard deviations where applicable.
Author Response

(The authors gave the same response as above.)

Round 2
Reviewer 2 Report
I can propose to accept the manuscript in its present form